# CONTEXT-INVARIANT, MULTI-VARIATE TIME SERIES REPRESENTATIONS

## ABSTRACT

Modern time series corpora, in particular those coming from sensor-based data, exhibit characteristics that have so far not been adequately addressed in the literature on representation learning for time series. In particular, such corpora often allow to distinguish between *exogenous* signals that describe a context which influences a given appliance and *endogenous* signals that describe the internal state of the appliance. We propose a temporal convolution network based embedding that improves on the state-of-the-art by incorporating recent advances in contrastive learning to the time series domain and by adopting a multi-resolution approach. Employing techniques borrowed from domain-adversarial learning, we achieve an invariance of the embeddings with respect to the context provided by the exogenous signal. To show the effectiveness of our approach, we contribute new data sets to the research community and use both new as well as existing data sets to empirically verify that we can separate normal from abnormal internal appliance behaviour independent of the external signals in data sets from IoT and DevOps.

## 1    INTRODUCTION

Many modern applications in the physical and virtual world are equipped with sensors that measure the state of the application, its sub-components, and the environment. Examples can be found in the Internet-of-Things (IoT) or in the DevOps/AIOPs space like monitoring wind turbines or cloud-based applications (Lu et al., 2009; Lohrmann & Kao, 2011; Nedelkoski et al., 2019; Li et al., 2020; Krupitzer et al., 2020). Leveraging such time series to identify abnormal appliance behaviour is appealing (see Figures 1b,1c for an overview of time series anomaly types), yet certain characteristics of these time series make them difficult to model with existing representation learning techniques.

First, time series corpora are often highly multi-variate as illustrated in Figure 1a. Each appliance has several sensors[1] associated with it that measure both *exogenous* signals from the environment as well as *endogenous* signals from the internal state of the appliance. Examples for exogenous variables include user behaviour/traffic in a web-based application or physical measurements such as temperature in an IoT context. Conversely, endogenous variables could include the CPU usage or the vibrations of a machine. Increased (application-internal) network traffic is expected with higher user load, and higher ambient temperatures naturally result in elevated temperature of a wind turbine. It is however important to understand when an application deviates from such expected patterns and exhibits unexpected behaviour relative to its environment. We call such effects *contextual anomalies*.

In addition, a defining characteristic of such time series corpora is the sparsity and noisiness of their associated labels. A label could indicate time spans when an application was in an a-typical state. This sparsity may be due to diverse reasons ranging from practical (e.g., data integration or cost of labelling) to fundamental (internal system failures may be exceedingly rare). Noisiness stems from the fact that failures are often subjective and human insight is needed or alarms come from rule-based systems that are themselves overly noisy (Bogatinovski et al., 2021; Wu & Keogh, 2021).

Hence, unsupervised or self-supervised representations of time series are needed that take the characteristics of such modern time series corpora into account. However, while the field of representation learning for sequential data has received considerable attention in domains s.a. natural language processing (NLP) (Lan et al., 2020; Mikolov et al., 2013; Fang et al., 2020; Jaiswal et al., 2021),

---

[1]Each of these sensors may also measure multiple statistics of the signal (e.g., min, max, avg, std).

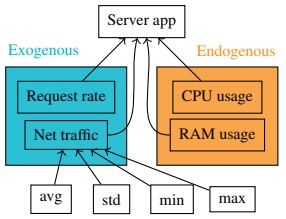 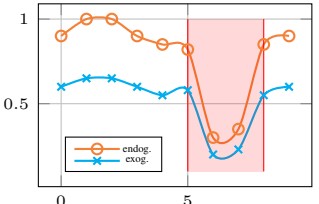 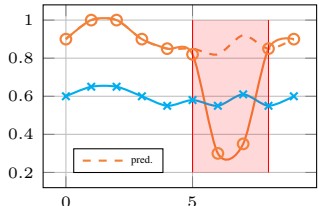

(a) **DevOps signal hierarchy example**. Each entity contains exogenous and endogenous sensors and each sensor is comprised of different summary statistics.

(b) **Pattern anomaly**: The time series is anomalous w.r.t its typical behaviour. The depency structure between exogenous and endogenous variables is preserved.

(c) **Contextual anomaly**: The time series is anomalous w.r.t context from another time series. The dependency structure between exogenous and endogenous variables is changed.

Figure 1: Structure of time series corpora (left) and different types of anomaly (middle and right).

similar work in the numerical time series domain remains rare. Specifically, rich, multi-purpose representations facilitating down-stream applications are common in NLP. Instead, feature extraction methods mostly dominate for time series (Lubba et al., 2019; Christ et al., 2017) with Franceschi et al. (2019) providing a notable exception based on temporal convolution networks (TCN). The main contribution of our paper is the extension of this TCN-based approach to cater for the aforementioned complications. We summarize our contributions as follows:

1. We propose *context-invariant* embeddings that allow to identify representations of time series that are invariant to the exogenous variables. We achieve this by adapting domain adversity (Ganin et al., 2016) to the time series domain.
2. We extend the TCN (Franceschi et al., 2019) model with (i) modern contrastive losses that we lift to the time series domain for the first time, (ii) data augmentation techniques, and (iii) considering time series simultaneously at multiple resolutions.[2]
3. We conduct an empirical study in which we show the effectiveness of our approach. We provide a semi-synthetic DevOps data set that we contribute to the research community and consider an under-explored wind turbine dataset (apart from classical synthetic and physical datasets).

Our quantitative results show that context-invariant embeddings indeed represent time series data such that contextual anomalies can be identified in a label-effective way. The qualitative results show that the embeddings allow us to navigate complex data sets in an explorative manner (e.g., considering nearest/farthest neighbours of interesting time series snippets).

## 2 REPRESENTATION LEARNING WITH CONTEXT INVARIANCE

To motivate our approach, consider a simplified system where under normal operation a single endogenous variable $y$ depends instantaneously on a single exogenous signal $x$ via a function $y = g(x) + \varepsilon$, where $\varepsilon$ is a noise term. The ideal signal to detect contextual anomalies (those that break this relation) is the residual $\delta := y - g(x)$. Under normal operation this signal carries no information about the exogenous variable and thresholding the magnitude of this residual signal can detect anomalies. Our approach is motivated by this setup but extends it to more complex situations where (i) exogenous & endogenous variables can be multivariate, (ii) the relation stochastic & highly non-linear, and (iii) may depend on the history of the system state. In this case we cannot simply compute a "residual signal", but instead we can try to learn unsupervised representations that are *invariant* to the exogenous variable. This means the embeddings should be independent of the driving signal as long as the endogenous variables respond in a typical manner, which captures some aspect of the residual signal of the toy example. In the following sections we formalize this intuition further and show that context invariance indeed helps detect such anomalies.

Let $Z = \{z_i \in D^T\}_{i=1}^N$ be a set of $N$ equally spaced time series $z_i$ of length at most $T \in \mathbb{N}$ where $D$ is a domain of numerical values. We do not assume time series to be of equal length. We assume

---

[2]Indeed, time series corpora typically consist of equally-spaced time series (e.g., time series with measurements in 1-min, 5-min or 10-min intervals). This allows us to reason at multiple resolutions.

a decomposition $Z = X \cup Y$ such that time series in $X$ allow to predict time series in $Y$. We call $X$ the set of environmental/exogenous time series and $Y$ the set of internal/endogenous time series. We assume that it is possible to predict $Y$ from $X$, but we make no assumption on causality.

The goal of this paper is to map sub-series of $Z$ into a high-dimensional embedding space $\mathbb{R}^M$ which preserves loosely defined properties such as: "normal" time series are close to each other and far away from "abnormal" states. This facilitates down-stream tasks such as time series classification or anomaly detection in a label sparse setting. In particular, our definition of "normal" should be *context-invariant*, that is, only changes in the dependency structure between $Y$ and $X$ should result in large distances in the embedding space. For these tasks, a limited number of labels is available that allows to identify a time span of abnormal behaviour. Typically, the amount of labels is such that a supervised approach is prohibitive and even evaluation may be a challenge.

Our representation learning approach consists of two main components: a *predictor network g* that ties the endogenous and exogenous time series together (either by predicting endogenous from exogenous variables, or vice-versa), and, an *embedding network f* which learns embeddings using contrastive losses. We can combine both in multiple ways. One extreme is a two-step approach where we learn embeddings on the residuals of the predictor network. The other extreme is an end-to-end approach, where we learn embeddings such that the distance between (multivariate) time series is adjusted based on the exogenous variables in a domain-adversarial way. Figure 2 depicts the main components of our approach. For the predictor network, we mainly resort to standard models, so we focus our exposition on the main (novel) components in the following.

## 2.1 Contrastive, self-supervised, learning of multi-resolution TCN network

The basic building block of our embedding network architecture (Franceschi et al., 2019) consists of stacked temporal dilated causal convolutions (Bai et al., 2018). We have multiple such networks, one per time resolution. We illustrate in Figure 4 the effect of aggregation on the input time series. To obtain a consolidated representation, the concatenated representations are mapped through a neural network. These multi-resolution representations allow the network to encode patterns that are more pronounced in the higher resolutions of the time series in a way that is more effective than an encoder which only operates on a single resolution. We choose resolutions manually as the natural granularities corresponding to the base frequency of the time series we consider in our empirical studies (e.g., seconds, minutes, hours).

Similar to (Franceschi et al., 2019), we rely on a contrastive, self-supervised learning approach to train the embedding network. This crucially relies on a *loss function* and a careful selection of positive $(a, b)_p \in (X, Y)$, reference $(c, d)_c \in (X, Y)$ and negative $(x, y)_n \in (X, Y)$ time series snippets on which to compute the loss terms (depicted in Figure 2). Similar time series should be close to each other and dissimilar time series distant from each other in the embedding space. For an embedding network $f_W$ with parameters $W$, the loss function takes the following general form:

$$\min_W \text{dist}(f_W((c,d)_c), f_W((a,b)_p)) + \max_W \text{dist}(f_W((x,y)_n, (a,b)_p)) \tag{1}$$

We choose $(a, b)_p, (c, d)_c$ such that $(c, d)_c \supseteq (a, b)_p$ while $(x, y)_n$ is such that $x \cap c \approx \emptyset, y \cap d \approx \emptyset$ (e.g., time snippets at different times and from different elements in the batch). Note further that $(x, y)_n$ is constructed to explicitly break the dependency structure in $Z$ by choosing $x$ to be the exogenous variables at a different time than $y$. During training, we further augment the examples randomly before feeding them to the TCN network. In particular we apply random jittering, scaling, flipping direction, 2d rotation around a center, permuting random segments, magnitude or time warping (Um et al., 2017) and window slicing or wrapping (Guennec et al., 2016).

Equation (1) is designed to support a variety of contrastive loss functions. Apart from the loss discussed in Franceschi et al. (2019), we rely on other, more recent losses which we describe in the following briefly. These losses, in particular the latter two, aim to avoid collapse of the embeddings while taking practical consideration (e.g., the size of the batch) into account.

The **SimCLR** (Chen et al., 2020) takes two random windows $z_A$ and $z_B$ of a time series and encodes it to get two representations $h_A$ and $h_B$. It then maximizes the similarity between these two representations from the same time series and dissimilarity between others representations in the batch using the Normalized Temperature-Scaled Cross-Entropy loss (Sohn, 2016) as the distance in (1).

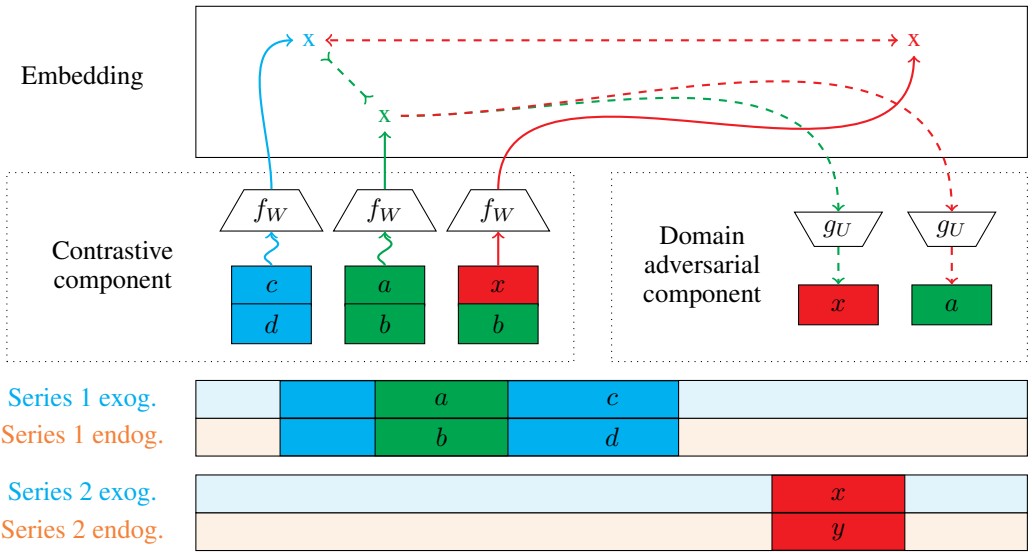

Figure 2: **Context-invariant embeddings with contrastive and domain adversarial learning.** We select negative samples such that the correlation structure between exogenous and endogenous signals is explicitly broken. For the other samples, we augment them randomly, denoted by the squiggly arrows. In addition to the contrastive component, we further add a domain adversarial component that, given a positive embedded sample, tries to reconstruct the context of the positive sample as badly as possible while reconstructing the context of the negative sample as well as possible.

Formally, for temperature parameter $\tau$ and mini-batch size of $N$ for each pair we define the loss as:

$$\ell_{A,B} = -\log \frac{\exp(\text{dot}(h_A, h_B)/\tau)}{\sum_{n=1,n\neq A}^{2N} \exp(\text{dot}(h_A, h_n)/\tau)}, \tag{2}$$

where $\text{dot}$ is the dot-product between $\ell_2$-normalized vectors. This contrastive loss benefits from larger $N$ which might not be feasible in the time series setting and thus we also explore other losses.

**Barlow Twins** (Zbontar et al., 2021) is a loss operating on two batches of different windows from the same respective time series embeddings, $Z_A$ and $Z_B$. It computes the cross-correlation matrix along the batch dimension and stores the result in a square matrix $C$. The final loss then encourages the diagonal terms in this matrix to be close to 1 and the off-diagonal terms be close to 0. Formally,

$$\ell = \sum_i (1 - C_{ii})^2 + \lambda \sum_i \sum_{i\neq j} C_{ij}^2, \tag{3}$$

where $\lambda > 0$ trades off the contribution of the first and second term in the loss. Intuitively, this decorrelation reduces the redundancy between the output embeddings forcing them to contain non-redundant information about the time series.

Unlike SimCLR, **MoCo** (He et al., 2020) uses *two* encoders to obtain representations for the two random windows from the same time series. The representations through the 2nd momentum encoder are preserved in a queue. During training, positive pairs in (2) are constructed from the current batch while negative pairs (denominator of (2) are constructed from the queue of embeddings. The 2nd encoder is updated by linear interpolation of the two encoder with a momentum-based moving average of their weights during training. By using a queue with a slowly changing encoder, this loss attempts to construct large and consistent embeddings which better samples the continuous high dimensional space, independent of the batch size.

## 2.2 COMBINATIONS OF PREDICTOR AND EMBEDDING NETWORK: FROM TRIVIAL TO CONTEXT-INVARIANT

We can combine the components described above to obtain context-invariant embeddings in a number of ways. The most trivial way is to ignore the predictor network $g_U$ and learn representations of

the entire set $Z$ (or similarly, only for the endogenous part $Y$). We denote this idea as `BasicEmb`. Next, a two-step approach for context-invariance consists of training the predictor network $g_U$ first, then learn embeddings $g_U(Y) - Y$ on top of the *residual* signals in a second step. We refer to this approach as `ResEmbRegr`. Finally, we describe how to combine both networks into an end-to-end model which we call `ContInvEmb`. This approach is the most flexible in the sense that it allows to adjust the strength of the context-invariance depending on the need of the application.

Our goal is to construct embeddings that are invariant to the corresponding (exogenous) context. Put differently, the embeddings should not contain information that allows prediction of the corresponding exogenous context. This in turn requires to regress the exogenous signals against the embedding but instead of minimizing the regression error, we attempt to maximize the regression error. More formally, we have

$$R(W) = \min_U \sum_{(a,b)_p,(x,y)_n} L_r(W, U, a, b, x, y) + \lambda \max_U \sum_{(a,b)_p} L_r(W, U, a, b), \qquad (4)$$

where $L_c(W, U, a, b, x, y)$ is a regression loss, $W$ are the parameters of the encoder network $f_W$, and $U$ are the free parameters of the prediction network $g_U$.

The loss in Equation (4) aims to reconstruct the exogenous variables from embeddings as badly as possible, thereby pushing embeddings to invariance wrt the exogenous context. The first loss term leads to good predictions of the exogenous variables shifted with respect to the embedded signals. The second term leads to bad predictions of exogenous variables $a$ from embeddings of positive examples $(a, b)_p$. Gradient reversal handles the adversarial learning aspect (Ganin et al., 2016).

Instead of using to a regression loss in (4), we resort to a multi-class classification problem by discretizing the input space. This has attractive properties in related tasks (Rabanser et al., 2020), but importantly for this particular application avoids explicitly handling trivial predictions (like $g(\cdot) = \pm\infty$) and domains are naturally bounded in the practical applications we consider. Combining (4) with a contrastive loss, we arrive at the following overall loss:

$$\min_W (L(W) + \lambda R(W)), \qquad (5)$$

where $L(W)$ is a contrastive loss as discussed in Section 2.1, and $R(W)$ is weighted by $\lambda$ (a hyper-paramater) and acts as a regularization term. For $\lambda > 0$, we obtain context-invariant embeddings and for $\lambda = 0$, we recover (Franceschi et al., 2019) (modulo our extensions).

## 3 RELATED WORK

Computer vision (CV) and NLP (Chen et al., 2020; van den Oord et al., 2019; He et al., 2020; Fang et al., 2020; Jaiswal et al., 2021) have embraced self-supervised representations. Most relevant to us, Sohn et al. (2021) learn representations with a contrastive loss to enable anomaly detection in CV. In contrast, time series analysis has not seen a similar adoption of self-supervised techniques for learning general-purpose representations. Franceschi et al. (2019) provide a notable exception by proposing a TCN based embedding (Bai et al., 2018) learnt with a contrastive loss.

The approach of Franceschi et al. (2019) departs from a rich field of feature extraction from time series (Lubba et al., 2019; Christ et al., 2017). While these approaches indeed classify time series well in practice, they mostly focus on the uni-variate case. Their extensions to the multi-variate case are out-performed by Franceschi et al. (2019); Bagnall et al. (2018). Furthermore, the versatility of the features learned by classical approaches is limited by the fact that distances in the induced embeddings are not properly learnt. We extend Franceschi et al. (2019) in the following directions: (i) we adopt it to be multi-resolution; (ii) we equip it with more recent contrastive loss functions and (iii) we turn it into a context-invariant embedding via domain-adversarial learning.

For (i), we note that the de facto choice in a multi-resolution context would be to sub-sample the time series. This is done in classical Wavelet analysis (Mallat, 1989). Instead, we draw inspiration from temporal hierarchical time series analysis (Athanasopoulos et al., 2017) and opt to aggregate time series along the time dimension leading to vectors with a fixed dimension. For (ii), we adopt loss functions (He et al., 2020; Zbontar et al., 2021; Chen et al., 2020) recently proposed for contrastive and self-supervised learning and transfer them to the time series domain. For (iii), we draw

on domain-adversarial representation learning, primarily Ganin et al. (2016). While Ganin et al. (2016) learns embeddings with respect to a specific label classification task, we instead adopt an unsupervised approach via contrastive learning. Further, we replace the domain classifier with an exogenous context regressor whose loss we seek to *maximize*. Such deep prediction networks can be sophisticated, as is the case in particular in the forecasting literature (see e.g., Benidis et al. (2020) for an overview). Our approach readily extends to these, but we restrict ourselves to standard neural regression models as they suffice in the scenarios that we consider for our empirical studies. Other approaches such as hypernetworks (Ha et al., 2016) are conceivable. Yet, they suffer from a lack of computational efficiency and robustness which inhibits their practical applicability.

## 4 EMPIRICAL EVALUATION

Our empirical evaluation consists of two parts. We first focus on dissecting the improvements on embedding learning through multi-resolution handling, contrastive losses and data-augmentation. Second, we examine the context-invariant representations towards their anomaly detection potential.

### 4.1 IMPROVEMENTS TO EMBEDDING

For this base experiment, we aim to show the versatility of the embeddings through a down-stream forecasting and classification task each of which we evaluate on two datasets representing the easy and the hard spectrum of the task. We do not aim for comprehensiveness in these first set of experiments but rather for an assessment of the relative improvements through our extensions. As a reference point to gauge the absolute accuracy better, we include a classical feature extraction baseline (Lubba et al., 2019), `Catch22`. In our experiments, we address the downstream classification and forecasting task via simple linear models.

In **classification**, we consider a synthetically generated data set for which we know the labels from the data generation process and the M5 forecasting competition dataset (Theodorou et al., 2021). The latter data set is a retail demand forecasting data set that has product categories associated with the time series. Using a (multinomial) logistic classifier, we aim to predict the labels in both data sets. The synthetic data set is designed such that a high classification accuracy can be and the labels are "objective". In contrast, for the M5 data set, a high classification accuracy cannot be expected. Apart from the amount of product categories available ($\approx$3k), time series associated to different products may have similar characteristics and hence, from a time series classification point of view, labels do not represent a ground truth, a common scenario in practice.

In **forecasting**, we predict electricity[3] and M5 (Makridakis & Spiliotis, 2021) using a shared linear forecaster. The forecaster takes the context embeddings from a large time series window together with a smaller context window of the actual time series to predict for the dataset's target horizon. This horizon is 24 time steps ahead for electricity and 28 days for M5 and compare the metrics with the corresponding test splits.

**Discussion.** Table 1 summarizes our findings. The more recent proposals for contrastive losses, `MoCo` and `Barlow Twins`, are superior, but there is no clear indication which of both losses is superior overall. While present, we remark that the relative improvements in accuracy are small when compared to the performance wins reported in the original domains for which these losses were designed. We speculate that this may be more attributable to the datasets in the respective domains (and the objectivity of the associated labels) than the loss functions themselves.

Note that including multi-resolution leads to such overwhelming improvements in the classification task that we show only multi-resolution embeddings (for the base resolution, factor 60 and 300) for the classification tasks as these are almost strictly superior ($> 10\%$). Similarly, we report results in the classification task with data augmentation, although improvements are not consistent for data augmentation (Appendix B.2 contains results for an ablation study). The effectiveness of multi-resolution may be surprising as higher capacity models in general and convolutions with a higher dilation in particular should in theory be able to model similar effects. However, adding multiple resolutions offers an inductive bias akin to lagged values in RNN-based forecasting models (e.g., Salinas et al. (2019)) which have been shown to lead to superior practical results.

---

[3]https://archive.ics.uci.edu/ml/datasets/ElectricityLoadDiagrams20112014

| Loss | Classification ↑: Syn | Classification ↑: M5 | Forecasting ↓ : Electricity | Forecasting ↓ : M5 |
|------|------|------|------|------|
| SimCLR | 97.00 | 1.34 | 0.097 | 0.70 |
| Barlow Twins | 97.25 | **5.93** | 0.091 | 0.699 |
| MoCo | **99.50** | 3.08 | **0.089** | **0.694** |
| Catch22 | 64.50 | 0.03 | 0.11 | 0.713 |
| Linear | - | - | 0.092 | 0.698 |
| DeepAR | - | - | 0.073 | 0.90 |

Table 1: Classification and forecasting accuracy using embeddings learnt with different losses. For forecasting, the accuracy metric shown is P50 loss. Results are averages over 5 runs. The last three lines are baselines using classic time series features, `Catch22`, and replacing embeddings with more historic values for the forecasting task, `Linear` and a pure forecasting method, `DeepAR`.

For forecasting, we note that past historic values instead of embeddings (`Linear` in Table 1) is a competitive approach, in particular for forecasting the M5 data set. Nevertheless, higher quality embeddings coincide with better forecasting accuracy (comparing `Catch22` with our embeddings). Compared with the Electricity dataset, the M5 dataset offers less overall structure so historic values offer a strong signal. Consequently, forecasting accuracy wins are more pronounced in the Electricity dataset.

## 4.2 CONTEXT INVARIANT EMBEDDINGS

In our main experiments, we consider the effect of domain-adversarially learnt context-invariant embeddings in both qualitative and quantitative experiments. The task consists of the identification of contextual anomalies. Perhaps surprisingly, qualitative evaluation is almost more meaningful for the anomaly detection task in general, and for contextual anomalies in particular, given the subjectivity and noisiness of the labels. Note that most publicly available anomaly detection data sets[4] are not suitable for the task, so we do not consider them. Instead, we evaluate on 4 different data sets, each of which allows for a separation into exogenous and endogenous signals. We discuss these data sets first, then the evaluation approach, the models under consideration, and finally discuss the results. Appendix B.3 contains further details.

**Datasets** The selection of the evaluation datasets aims to balance physical and virtual appliances as well as synthetic and real-world data. Note that synthetic data come with perfect labels, while real-world data typically does not. While lamentable, we believe that this subjectivity and noisiness must be embraced as fundamental in the task. Appendix A contains further details on the data sets.

*Synthetic data.* We generate a total of $4 \times 360$ time series of length 700, based on simple generative models. We generate two exogenous signals, as well as two endogenous signals. We inject two types of anomalies into the data: (i) *pattern anomalies*, i.e., anomalies in the exogenous which are also instantly reflected in the endogenous variables and (ii) *contextual* anomalies only in the endogenous variables. We aim to detect contextual anomalies.

*Pendulum.* We consider the case of a swinging pendulum with added control signals, where we control the dampening of the acceleration from the outside as an exogenous signal (towards which we want to be invariant) and consider as contextual anomamlies those where we inject an anomalies as a change in the length of the chord which we capture as part of the endogenous signal. Our aim with this data set is to understand how well our models handle cases where the dependency structure between $X$ and $Y$ is more complex.

*DevOps.* This is a new, semi-synthetic data set[5] that we generated for the purpose of this publication to resemble commonly observed data sets behind corporate firewalls. The object under consideration is a popular cloud-based microservice demo application,[6] which is commonly used in an AIOPs context (Wu et al., 2020). As exogenous signal, we record user interaction approximated by the net-

---

[4]Wu & Keogh (2021) convincingly argue that many of these datasets should be abandoned.

[5]Jointly with this publication, we open-source both the raw data as well as the set-up to produce the data. To the best of our knowledge, we are the first to extract a data set from this set-up that allows the machine learning community to interact with this area without deep engineering knowledge which is more present in the system's community where the data generation framework is typically considered.

[6]`https://github.com/microservices-demo/microservices-demo`

work outbound traffic of an application that induces synthetic load on the application and which we fully control. The endogenous signals consists of metrics like CPU, memory and others of the actual microservice application. Each recorded metric has multiple statistics available. We inject anomalies both in the user behavior (leading to pattern anomalies) and in the internal state (contextual anomalies). We want to ignore the former and find the latter. The appendix contains illustrative plots similar to Figure 6. Note that although we have almost total control of the application and its anomalies, the labelling of anomalies is still not perfect, thereby adding further to the complications of public anomaly detection benchmarks (Wu & Keogh, 2021).

*Turbine.* We consider a wind turbine data set open-sourced[7] by Energias de Portugal. The time series panel can be separated into exogenous and endogenous signals. The former consists of wind speed/direction, ambient temperature and, the pitch angle of the blades (this is controlled from the outside and including it improves the quality of the predictive model). The latter consists of rotation speeds for the turbine and generators, internal temperature on different components, as well as the power output each. All series are available for 4 distinct turbines, are sampled at a 10 minute frequency and are available for 2016. Note that this data set contains only few (43) labelled anomalies and visual inspections of the data reveals inconsistencies with these labels, e.g., some time series appear to be mislabeled (see Sec. 4.2 for an example). The providence of these labels is from automated alarming systems which are often threshold based. Hence, quantitative evaluations cannot be taken at face value. However, given the rich structure of the data and the fuzziness of the task stemming from the labels, the versatility of our approach can be illustrated qualitatively.

**Models & Evaluation**   We evaluate the following model configurations for their suitability for the contextual anomaly detection task: `BasicEmb`, the modified (Franceschi et al., 2019) (with multi-resolution) learnt ignoring the structure imposed by endogenous and exogenous signals; `ResEmbRegr`, embeddings on the residuals of a predictive models (a feed-forward neural network); and `ContInvEmb`, context-invariant embeddings with a simple linear model for the prediction task (the simplicity ensure that the embeddings are adjusted enough). Moreover, we also provide results for three baseline approaches: `ResTresh`, which computes residuals as in `ResEmbRegr` combined with a simple thresholding mechanism; and two instances of (Lubba et al., 2019): `Catch22` and and `ResCatch22` which compute a feature vector per original and residual time series respecitvely similar to `ResEmbRegr`. The only hyperparameter tuning that we perform is on the Synthetic data set as we do not have enough labels available otherwise.

For all methods, we report standard AUROC scores on the contextual anomaly detection task. For the `ResThresh` method, we compute the AUROC score based on the maximum residual value over the full residual series. For the embedding-based approaches, we use a $k$-nearest-neighbor classifier in the embedding space to determine a discrete anomaly label.

**Quantiative Results**   Tables 2 summarizes the quantiative results. First, we note that the difficulty of contextual anomaly detection differs widely with the data sets. For example, contextual anomaly detection seems relatively easier on the Turbine data set compared with the DevOps data set despite the latter being semi-synthetically generated with controlled labels. Second, we note that `ContInvEmb` leads to overall superior results when comparing the embedding based approaches (the first three columns in Table 2) and is overall competitive results in many cases, but not always. For the DevOps and Turbine data sets `ResTresh` is overall best by a margin. One explanation is that `ResTresh` best resembles the label generation process by the automated alarming systems on the turbines. For Pendulum data set it is worthwhile to note that `ContInvEmb` delivers an overall superior approach despite the complex non-linear interaction between the exogenous and endogenous signals and only a linear context-predictor component.

**Qualitative Results: a case study on wind turbines**   The embeddings allow to navigate the time series corpus via distances which can be helpful in exploring the data set and uncovering data or label issues. In Figure 3 we show this in qualitative results for context-invariant embeddings. The first two columns show sanity checks: as expected, reference time series (in the top row) labelled as normal or abnormal have as their nearest neighbors (in the same column as the reference time series below it) normal and abnormal time series respectively. Despite the multi-variate nature of the data, visual inspection confirms that the nearest neighbors are plausible. The last column in Figure 3

---

[7]https://opendata.edp.com/pages/homepage/

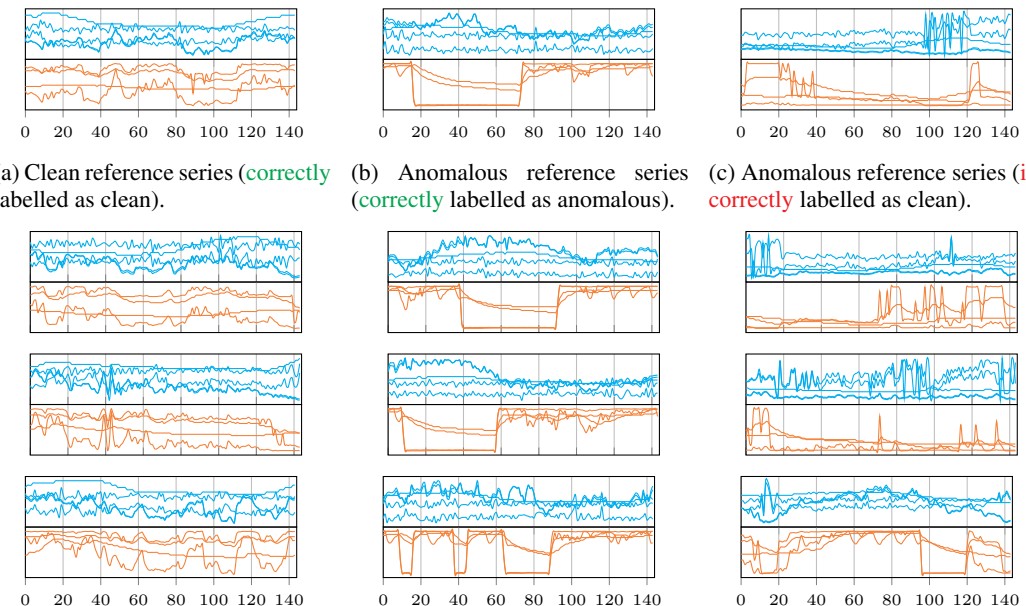

Figure 3: **Nearest-neighbor analysis of context-invariant approach on the Turbine dataset.** Each panel shows exogenous signals on the top and endogenous signals at the bottom. The first row shows (a) a normal reference series; (b) an abnormal reference series (anomaly between time steps 20 and 70); and (c) an anomalous reference series that was mistakenly labeled as clean. Below each of these series we show 3 nearest neighbors in the learned embedding space. For the left and center column, both the correct label and our predicted label match. For series (c), the reference time series is labelled as clean, but its nearest neighbors are abnormal. It is apparent that the erratic spiking patterns in either exogenous/endogenous signals are usually not reflected in the other series.

|           | BasicEmb            | ResEmbRegr          | ContInvEmb          | ResTresh            | Catch22             | ResCatch22          |
|-----------|---------------------|---------------------|---------------------|---------------------|---------------------|---------------------|
| Synthetic | 0.512 ($\pm$ 0.022) | **1.000** ($\pm$ 0.000) | 0.999 ($\pm$ 0.002) | **1.000** ($\pm$ 0.000) | 0.494 ($\pm$ 0.008) | **1.000** ($\pm$ 0.000) |
| Pendulum  | 0.969 ($\pm$ 0.013) | 0.951 ($\pm$ 0.015) | **0.980** ($\pm$ 0.002) | 0.510 ($\pm$ 0.000) | 0.904 ($\pm$ 0.000) | 0.891 ($\pm$ 0.000) |
| DevOps    | 0.535 ($\pm$ 0.041) | 0.532 ($\pm$ 0.036) | 0.587 ($\pm$ 0.007) | **0.619** ($\pm$ 0.000) | 0.573 ($\pm$ 0.000) | 0.573 ($\pm$ 0.000) |
| Turbine   | 0.632 ($\pm$ 0.015) | 0.725 ($\pm$ 0.018) | 0.736 ($\pm$ 0.022) | **0.845** ($\pm$ 0.000) | 0.512 ($\pm$ 0.000) | 0.680 ($\pm$ 0.000) |

Table 2: AUROC results on the anomaly detection task with 5 seeds. Larger values are better.

shows an example where a reference time series is labelled as normal, but its nearest neighbors consist of time series that are labeled as abnormal. This could point to an issues with the labels. In this case, without further domain knowledge, it may make sense to re-label the reference time series in column (c) as abnormal for the abnormal stretches 0-40 and 100-130.

## 5 CONCLUSION

In this work, we presented self-supervised learning of time series embeddings that are invariant with respect to a known and fully-observed context. While the architectures that we presented here lean on techniques invented for computer vision, we make non-trivial contributions to adapt them to the time series domain. For example, we equip our embeddings to consider multiple resolutions of the original sensor signals simultaneously. We observe that the learned embeddings are sensitive to changes of the dependency structure between exogenous and endogenous variables. As confirmed in our evaluation, this allows our approach to learn embeddings that separate dependency-breaking anomalies in the state of the appliance which is the object of interest.

Potential future works could explore techniques from causal discovery (Haufe et al., 2009; Qiu et al., 2020; 2012) to automatically derive an exogenous/endogenous decomposition of the multi-variate time series panel and extensions to causal representation learning (Schölkopf et al., 2021).

## 6 REPRODUCIBILITY

We use a combination of public datasets and synthetic data sets generated using publicy available code. Method implementations will be open sourced as part of the review process. Finally, we will share the data sets, the models to create the synthetic data and the code with notebooks that allows to reproduce all our results with the final version of this paper.

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
