# OpenReview forum: "Context-invariant, multi-variate time series representations"
_ICLR.cc/2022/Conference — ICLR 2022 Submitted_

### Official Review · Reviewer_N1hN · 2021-11-02

**Correctness:** 3
**Technical Novelty And Significance:** 3
**Empirical Novelty And Significance:** 2
**Recommendation:** 5
**Confidence:** 3

**Main Review:**

This paper adapts self-supervised learning to the time series analysis domain for the anomaly detection problem. Authors propose a contrastive component to distinguish similar/dissimilar time sires and a domain adversarial component to ensure the embeddings are context invariant.
Generally, I think the studied problem is relatively new and the model design contains novel ideas. However, I also have the following concerns:
1. Although the overall idea and the whole concepts of each component are very clear, some details of the proposed method are not easy to get. For example:
* What is one training example feed into the network ($N \times T$ matrix contains both internal and external series or just one series?)?
* How to deal with the input with different resolutions?
* What is the extract calculation of $L_r(W, U, a, b, x, y)$ in Eq. 4? What does it mean “The first loss term leads to good predictions of the exogenous variables shifted with respect to the embedded signals”?
2. The justification for identifying representations of time series that are invariant to the exogenous variables is not enough. Especially, why this is important and necessary?
3. While the experimental results seem good, they are not very strong and supportive to the motivation. First, the experimental results are not better than the baseline supervised methods. I know it’s reasonable since you do not use labels to train the embedding part, but the classifier is still trained with labels. For these small-scale datasets, I doubt the representative ability of embeddings since the classifier may influence the performance hugely. Second, the most attractive point of self-supervised learning is the transferability of learned features, which is not evaluated in this paper. Unsupervised learning features and then supervised learning the classifier can not support the method for anomaly detection applications.


**Summary Of The Paper:**

This paper adapts self-supervised learning to the time series analysis domain for the anomaly detection problem. Authors propose a contrastive component to distinguish similar/dissimilar time sires and a domain adversarial component to ensure the embeddings are context invariant.

**Summary Of The Review:**

The paper studies a relatively new topic and the design contains new ideas in the specific area. However, some details of the model are not presented clearly and the experimental design could be improved.

---

> ### Author Response · Authors · 2021-11-16
> **Response 3.2**
>
> > The justification for identifying representations of time series that are invariant to the exogenous variables is not enough. Especially, why is this important and necessary?
>
> Traditional time series anomaly detection methods have mostly focussed on identifying pattern anomalies, i.e. unusual/unseen behaviour with respect to past behaviour. In our work, we focus on contextual anomalies, i.e. identifying changes in the typical correlation structure between multivariate time series signals. In a wide variety of scenarios, such contextual anomalies remain either unnoticed or we cannot distinguish them from pattern anomalies.
> To illustrate the practical relevance of the problem of identifying contextual anomalies, we present two examples:
> - *Wind turbine*: Assume a wind turbine scenario in which we have access to both a wind speed sensor (exogenous) and a rotational speed sensor (endogenous). Exemplary pattern anomaly: We observe a sudden drop in both the wind speed and the rotational speed of the wind turbine. This can happen because of changing weather conditions. Exemplary contextual anomaly: We observe a sudden drop in the rotational speed but the wind speed stays constant. This can happen because of a sudden mechanical failure of the turbine. See linked picture for an example of both types of anomalies in the wind turbine data set: https://ibb.co/TtbXGTF
> - *DevOps*: Assume a DevOps scenario in which we have access to a user traffic sensor (exogenous) and both a CPU usage and a RAM usage sensor (endogenous). Exemplary pattern anomaly: We observe a sudden spike in active users, CPU usage, and memory all at the same time. This can happen because of a popular out-of-stock item suddenly being on sale again at an online store. Exemplary contextual anomaly: We observe a sudden spike in CPU and RAM usage while the active user count stays constant. This can happen because of a malicious program trying to use system resources.
>
> > While the experimental results seem good, they are not very strong and supportive to the motivation. First, the experimental results are not better than the baseline supervised methods. I know it’s reasonable since you do not use labels to train the embedding part, but the classifier is still trained with labels. For these small-scale datasets, I doubt the representative ability of embeddings since the classifier may influence the performance hugely. Second, the most attractive point of self-supervised learning is the transferability of learned features, which is not evaluated in this paper. Unsupervised learning features and then supervised learning the classifier can not support the method for anomaly detection applications.
>
> Regarding the first point, we clarify that we do not make use of any label information for training of our proposed method. Labels are purely used for evaluation purposes to determine the correctness of the anomaly classification outcome. For embedding-based approaches, this involves a nearest neighbor search in the embedding space and taking a majority vote of the returned neighbors’ anomaly labels. For the second point, we agree with the reviewer that indeed, transferability is an important point of features learnt in a self-supervised fashion. In our situation, this cannot be straight-forwardly applied as the domain adversarial part highly depends on the data set as is evident in the data sets. For example, the relation between the exogenous and the endogenous variables is different in pendulum (highly complex) and in the fully synthetic data sets (less complex). It would be an interesting (and non-trivial) avenue of future work to apply the domain adversarial part to pre-trained Francheschi et. al embeddings. In the current manuscript, we transfer hyperparameters.

---

> ### Author Response · Authors · 2021-11-16
> **Response 3.1**
>
> We thank the reviewer for their comments. We address individual concerns below:
>
> > What is one training example fed into the network (N x T matrix contains both internal and external series or just one series?)?
>
> In each training step, a matrix of $N \times Z \times T$ is fed to the model. $N$ is the batch size, $Z$ is the number of multivariate channels and $T$ is the number of time steps. Recall that $Z = X \cup Y$, i.e. we can decompose $Z$ into exogenous variables $X$ and endogenous variables $Y$. This decomposition is crucial as it allows our method to identify the underlying correlation structure between the two types of variables. Training happens in batches of fixed size, in our case 16. At inference time, a variable amount of samples can be fed into the model. We will clarify this in the manuscript.
>
> > How to deal with the input with different resolutions?
>
> We are not entirely sure what the exact question is by the reviewer and kindly ask the reviewer to clarify.
> We note that we assume that the data set that we consider consists of time series with the same resolution (e.g. seconds). This means that we do not handle datasets that consist of multiple, different frequencies yet. However, based off the base resolution, we go to a different resolutions by choosing an aggregation operation (mean, sum, max, min) and a window size (e.g. 60 to go from seconds to minutes; we have made sure that we can handle irregular aggregations such as months) and aggregate the original input in that fashion. Concretely, we only aggregate over the time dimension $T$ while keeping $N$ and $C$ constant.
>
> > What is the exact calculation of L_r(W,U,a,b,x,y) in Eq. 4? What does it mean “The first loss term leads to good predictions of the exogenous variables shifted with respect to the embedded signals”?
>
> Reviewer PwXK also asked for this and we added a clarification of this. $L_r$ is the negative log likelihood that we both maximize or minimize, depending on whether we’re using a positive or negative example. We added the following clarification:
>
> The loss in Equation (4) aims to reconstruct the exogenous variables from embeddings as badly as possible, thereby pushing embeddings to invariance wrt the exogenous context.
> In more detail, the right-most term in (4) with a $\min$ instead of a $\max$ is a reconstruction loss. For an embedded positive example, it would aim at reconstructing the positive example as well as possible. Here, we choose maximization instead of minimization, so we aim to reconstruct the example as badly as possible. While other losses such as an $\ell_2$ loss are possible, we opt for the following realization
> \begin{equation*}
>     L_r (W, U, a, b) = - \log p( a| a,b, U, W ) \,
> \end{equation*}
> that is, the negative log likelihood in a multinomial classification, where we have discretized the continuous $(a,b)$ accordingly and $g_U ( f_W ( (a,b)))$ provides an estimate of $a$. For the left-hand term in (4), the negative log likelihood would be
> \begin{equation*}
>     L_r (W, U, a, b, x) = - \log p( x | a,b, W, U ) \,
> \end{equation*}
> with $g_U ( f_W ( (a,b)))$ providing an estimate of $x$.

---

### Official Review · Reviewer_PwXK · 2021-11-02

**Correctness:** 2
**Technical Novelty And Significance:** 3
**Empirical Novelty And Significance:** 2
**Recommendation:** 3
**Confidence:** 3

**Main Review:**

Pros

+ interesting and relatively unexplored issue in the literature

Cons

- article needs deep proofreading

- experimental part not clear enough. Mix in the datasets between experiments is confusing.

====

The article contains a number of English mistakes and deserves a thorough proofreading.

Figure 2 is the weak point of the article: whereas it is suppose to give a general overview of the system, it not clear for me.

dot should be removed from (2) and bolded font should be used for vectors to make the reading more convenient.

typo: in (4) there is a confusion between Lc and Lr
more generally, the formalization of section 2.2 could be improved.
The loss function R is not given in (5) whereas it is supposed to be an important contribution of the paper.

The classification tasks of section 4.1 are not clear for me: I don't understand the introduction of the synthetic dataset and I don't see the interest of classifying a global signal wrt the introduced architecture.
Some additional content in appendix (& even in the body of the article) are required.

Regarding the forcasting, we don't know the horizon of the prediction and we have no information about the number of scales taken into account. This information is critical both for understanding and reproduction.

In section 4.2, what is the discretization step? I don't know how to interpret results without this key information.
The proposed approaches should also be compared with a simple baseline to provide a better demonstration of the interest of the method.

At least, the authors should build a synthetic dataset that enables us to understand the strengths and weaknesses of the different formulations.


**Summary Of The Paper:**

This article deals with representation of multivariate signals in the case where we can distinguish endogenous & exogenous channels. This article mainly relies on (Franceschi et al.,2019), whose proposal consists in representing signals using Dilated-TCNN and to apply triplet loss on the sequences as in word2vec. Then the authors introduce 3 cases: normal behavior, anomalies where both endogenous & exogenous are impacted and partial anomalies.
The encoded representation of a contextualized signal should be close to its temporal neigbors.
Then, the authors tackle prediction & classification tasks on M5 and electricity datasets. The experimental part is a little bit confusing, as the datasets used in the different experiments are not the same.
Generally speaking, results are very difficult to interpret.

**Summary Of The Review:**

This work seems very promising but the interest of the invariant representation is not clearly demonstrated according to me. Some experiments and critical information are missing so that I have to reject this paper. I strongly encourage the author to write & submit an improved version of this article in another conference.

---

> ### Author Response · Authors · 2021-11-16
> **Response 2.3**
>
> > Regarding the forecasting, we don't know the horizon of the prediction and we have no information about the number of scales taken into account. This information is critical both for understanding and reproduction.
>
> We have added detailed information on the forecasting task in the Appendix. This now includes (i) the sizes of the datasets as well as the sizes of the context windows; (ii) the size of the linear model; as well as (iii) the training hyperparameters. For the electricity data set we use a 24-step forecasting horizon and for the M5 data set we use a 28-days forecasting horizon. Table 4 in the appendix now contains more details on this setup. We are not sure what the reviewer meant with the “number of scales” and kindly ask for clarification. For the forecasting task, we refrained from using aggregation in the time dimension.
>
> > In section 4.2, what is the discretization step? I don't know how to interpret results without this key information. The proposed approaches should also be compared with a simple baseline to provide a better demonstration of the interest of the method.
>
> We agree with the reviewer and also reviewer krgP that more baselines are important to gauge the difficulty of the overall task. We have started work to include a deep learning based and a classical time series anomaly detection baselines.
> For the discretization step, we apologize for leaving out this indeed important detail. We have chosen a discretization of 20 buckets where we choose the buckets using quantiles per time series. This provides us with a natural normalization of our data set.
> We have since added further baselines (OmniAnomaly, Local Outlier Factor, Isolation Forest, One-class SVM), see the answer for reviewer krgP [here](https://openreview.net/forum?id=7sz69eztw9&noteId=8lQF6Eso3Mz) for more details.
>
> > At least, the authors should build a synthetic dataset that enables us to understand the strengths and weaknesses of the different formulations.
>
> As part of our evaluation, we provide one fully synthetic data set, created with a simple generative model, and a second physics-inspired data set, the pendulum data set, which is intended to serve the purpose that the reviewer asked for. We understand however that this may have been buried in the exposition of the experiments and also in the way we evaluate our baselines and our proposed method. Hence, we will distinguish in our evaluations now between contextual and pattern anomalies and report these results here, once we have obtained them.

---

> > ### Author Response · Authors · 2021-11-23
> > **More evaluations on pattern anomalies and contextual anomalies**
> >
> > As promised, we are providing more results on the identifiability of pattern anomalies and contextual anomalies. For the synthetic dataset, we create two distinct labelings:
> > 1) A _2-class_ labeling where both non-anomalous series and pattern-anomalous series are assigned the label 0 and contextual anomalies only are assigned the label 1; and
> > 2) A _3-class_ labeling where non-anomalous series are labeled as 0, pattern anomalies are labeled as 1, and contextual anomalies are labeled as 2.
> >
> > Performance on the first labeling indicates how well each method is capable of separating contextual anomalies from all encounterable scenarios (1-vs-all setting) while performance on the second labeling indicates the level of disentanglement of individual anomaly types in the embedding space. In the following table, we report the weighted F-1 scores for both labeling strategies.
> >
> > | Labeling \ Method | Context-invariant embedding |  Basic embedding  | Residual embedding |      Catch22      |  Residual Catch22 |
> > |:-----------------:|:---------------------------:|:-----------------:|:------------------:|:-----------------:|:-----------------:|
> > |      2-class      |      0.984 (+/- 0.012)      | 0.851 (+/- 0.037) |  0.983 (+/- 0.010) | 0.854 (+/- 0.019) | 0.989 (+/- 0.008) |
> > |      3-class      |      0.887 (+/- 0.029)      | 0.736 (+/- 0.033) |  0.594 (+/- 0.075) | 0.708 (+/- 0.026) | 0.644 (+/- 0.058) |
> >
> > We observe that context-invariant embeddings provide both (i) strong 1-vs-all performance (2-class) and at the same time still (ii) enable superior identification of non-anomalous and pattern-anomalous series (3-class). We also find that residual-based approaches (i.e. Residual Embedding and Residual Catch22) provide strong 1-vs-all performance but they do so at the expense of pattern anomaly identification. These findings suggests that context-invariant embeddings enable us to learn more complex dependence structures and patterns that go beyond simple residual signals.

---

> > > ### Comment · Reviewer_PwXK · 2021-11-29
> > > **In depth paper modifications**
> > >
> > > The authors propose a lot of improvement on their paper. However, it corresponds to so many changes that it modifies the original contribution: I still think that this work is promising but the new version of this article has to be submitted in another conference.

---

> ### Author Response · Authors · 2021-11-16
> **Response 2.2**
>
> > Figure 2 is the weak point of the article: whereas it is suppose to give a general overview of the system, it not clear for me.
>
> Based on the feedback by the reviewer, we have reworked the figure to more clearly separate the two parts of the figure: (i) how to select positive/negative examples for the contrastive loss and (ii) how the neural network architecture and loss can be visualized.
>
> We further want to add the following explanation which we hope will add some further clarification. Given a reference series from the training set and it’s decomposition into exogenous (Series 1 exog.) and endogenous (Series 1 endog.) signals, we first select a context frame (highlighted in blue). The respective exogenous context frame is given by $c$ and the endogenous context frame is given by $d$. Within this context frame, we select a positive frame (in green) where the exogenous subframe is denoted $a$ and the endogenous subframe is denoted $b$. For our negative example, we pick the exogenous frame within another random time series in the batch (Series 2 exog.) and take $y=b$ (i.e. the endogenous frame from the positive sample) as the endogenous frame (in red). This deliberate choice of composing our negative sample ensures that the dependency structure between the exogenous and endogenous variables in our negative example is explicitly broken. Having constructed these samples, our method then uses both a contrastive component  and a domain adversarial component to train a context-invariant encoder $f_W$. The contrastive component tries to minimize the distance of the context sample and the positive sample, while at the same time trying to maximize the distance between the context sample and the negative sample. At the same time, the domain-adversarial component tries to reconstruct the positive sample’s exogenous component $a$ as badly as possible. This ensures that the representation of the positive sample contains as little information as possible about its original context given by the exogenous variables. Moreover, the domain-adversarial component also tries to reconstruct an arbitrary exogenous signal provided by the exogenous negative sample $x$ as well as possible. This further helps in reducing context information from the positive sample.
>
> In contrast to the original version of Figure 2, we performed a few slight tweaks to fix a notational disconnect between Figure 2 and Section 2.1 and separated the sample selection and architecture overview into two distinct subfigures. Find the updated version here: https://ibb.co/Zgmc3bQ
>
> > The classification tasks of section 4.1 are not clear for me: I don't understand the introduction of the synthetic dataset and I don't see the interest of classifying a global signal wrt the introduced architecture. Some additional content in appendix (& even in the body of the article) are required.
>
> Taking your feedback into account, we have hence clarified the classification task: for the synthetic data set, the labels correspond to the type of modulation that we applied to a simple base generative model (sinusoidal data). The modulations are (1) amplitude modulation, (2) linear increase and decrease of the mean over the entire time series, (3) shifted mean for a random subseries and (4) the identity. Figure 12 in the appendix contains an illustration. Amplitudes, magnitude of shifts and beginning of phase are randomly sampled for each time series.
> For the M5 data set, the labels correspond to the product categories associated with the product categories.
>
> We note that the experiments in 4.1 do not contain a domain adversarial part. The goal of the classification and forecasting experiments is to show-case the improvements via different contrastive losses and the aggregation operations that we introduce. We use this type of indirect evaluation (via improvements in the forecasting/classification downstream tasks) because a direct evaluation of the contrastive losses themselves without a downstream task is harder.

---

> ### Author Response · Authors · 2021-11-16
> **Response 2.1**
>
> We thank the reviewer for their comments. We address individual concerns below:
>
> > The experimental part is a little bit confusing, as the datasets used in the different experiments are not the same. Generally speaking, results are very difficult to interpret.
>
> We agree with the reviewer that the exposition can be improved and clarified. We have added some missing details already in the first part of the experiments that hopefully add some clarification already (see further below). We further added more baselines for the second part of the experiments which allow to better contextualize our methods in light of more standard anomaly detection methods. See answer for reviewer krgP [here](https://openreview.net/forum?id=7sz69eztw9&noteId=8lQF6Eso3Mz).
>
> We will continue to work on this and update the reviewer as we make changes.
>
> > The article contains a number of English mistakes and deserves a thorough proofreading.
> > dot should be removed from (2) and bolded font should be used for vectors to make the reading more convenient.
> > typo: in (4) there is a confusion between Lc and Lr more generally, the formalization of section 2.2 could be improved. The loss function R is not given in (5) whereas it is supposed to be an important contribution of the paper.
>
> We thank the reviewer for their attention to detail and will make sure that the updated revision is thoroughly proofread.
> For the comment regarding equation (4), this is indeed a mistake in the writing. The formula is correct, but in the ensuing text, it should have been $L_r$ instead of $L_c$. Thank you for pointing this out.
> For equation (5), we have the loss function $R$ defined in equation (4). We had not further specified the constituent of equation (4) in our original manuscript as we believe these to be standard (an $\ell_2$ loss or a negative log likelihood of a multinomial classification). The other part of the loss (equation (5)) is given by contrastive losses which we spell out in detail in Section 2.1. Since another reviewer also asked for clarification of this section, we changed it in the following way. If this is still too vague, we are more than happy to iterate over this more.
>
> The loss in Equation (4) aims to reconstruct the exogenous variables from embeddings as badly as possible, thereby pushing embeddings to  invariance wrt the exogenous context.
> In more detail, the right-most term in (4) with a $\min$ instead of a $\max$ is a reconstruction loss. For an embedded positive example, it would aim at reconstructing the positive example as well as possible. Here, we choose maximization instead of minimization, so we aim to reconstruct the example as badly as possible. While other losses such as an $\ell_2$ loss are possible, we opt for the following realization
> \begin{equation*}
>     L_r (W, U, a, b) = - \log p( a| a,b, U, W ) \,
> \end{equation*}
> that is, the negative log likelihood in a multinomial classification, where we have discretized the continuous $(a,b)$ accordingly and $g_U ( f_W ( (a,b)))$ provides an estimate of $a$. For the left-hand term in (4), the negative log likelihood would be
> \begin{equation*}
>     L_r (W, U, a, b, x) = - \log p( x | a,b, W, U ) \,
> \end{equation*}
> with $g_U ( f_W ( (a,b)))$ providing an estimate of $x$.

---

### Official Review · Reviewer_krgP · 2021-11-05

**Correctness:** 3
**Technical Novelty And Significance:** 2
**Empirical Novelty And Significance:** 2
**Recommendation:** 3
**Confidence:** 4

**Main Review:**

Pros:
* The article is well written and easy to follow.
* The model is elegant, well described.
* The approach and the motivation are clearly stated.

Cons:
* The experimental part is very light and does not convince me of the interest of the proposed approach. No other deep learning models are used as baseline (even if not adapted to the context-invariant setting) when there is a rich literature about anomaly detection in time series. I understand that the goal is not to evaluate the performances in classification/anomaly detection but the learned representation, but I can not judge the difficulty of the used datasets with the provided baselines.
* The novelty of the paper is not great wrt the previous work, it consists mainly of the use of adversarial learning.

**Summary Of The Paper:**

The paper presents a model for learning representations of time series. It addresses, in particular, the context of multivariate time series having exogenous and endogenous dimensions, and where the goal is to learn good representations independently from the context (exogenous dimensions). This work is an extension of existing work, the additions being the multivariate aspect and the context-invariant aspect. The context-invariant problem is treated thanks to an adversarial learning mechanism.  Experiments are conducted on synthetic and real datasets.

**Summary Of The Review:**

The addressed problem is interesting and the proposed model is elegant but I think that the experimental part is too weak to assess the quality of the proposed approach and that the novelty is not sufficient wrt previous works.

---

> ### Author Response · Authors · 2021-11-16
> **Response**
>
> We thank the reviewer for their comments. We address individual concerns below:
>
> > The experimental part is very light and does not convince me of the interest of the proposed approach. No other deep learning models are used as baseline (even if not adapted to the context-invariant setting) when there is a rich literature about anomaly detection in time series. I understand that the goal is not to evaluate the performances in classification/anomaly detection but the learned representation, but I can not judge the difficulty of the used datasets with the provided baselines.
>
> We hadn’t included more anomaly detection baselines for the reasons that the reviewer listed. However, you and reviewer PwXK, convinced us that this is indeed necessary. We are therefore including a deep learning baseline (OmniAnomaly) and classical anomaly detection baselines (Local Outlier Factor, Isolation Forest, One-class SVM). The AUROC results are summarized in the table below:
>
> | Dataset \ Methods | Local Outlier Factor |  Isolation Forest |   One-class SVM   |    OmniAnomaly    | Context invariant embeddings |
> |:-----------------:|:--------------------:|:-----------------:|:-----------------:|:-----------------:|:----------------------------:|
> |     Synthetic     |   0.557 (+/- 0.016)  | 0.500 (+/- 0.000) | 0.499 (+/- 0.000) | 0.788 (+/- 0.108) |       0.999 (+/- 0.002)      |
> |      Pendulum     |   0.541 (+/- 0.000)  | 0.502 (+/- 0.000) | 0.500 (+/- 0.000) | 0.642 (+/- 0.082) |       0.980 (+/- 0.002)      |
> |       DevOps      |   0.518 (+/- 0.000)  | 0.489 (+/- 0.000) | 0.569 (+/- 0.000) | 0.513 (+/- 0.023) |       0.587 (+/- 0.007)      |
> |      Turbine      |   0.578 (+/- 0.000)  | 0.510 (+/- 0.000) | 0.521 (+/- 0.000) | 0.561 (+/- 0.026) |       0.736 (+/- 0.022)      |
>
> In contrast to context-invariant embeddings, both classical approaches and OmniAnomaly fail at consistently identifying anomalous changes in the dependence structure. Note that the above experiments are carried out on the original data set and we will add results for those methods operating on residuals in the next days.
>
> > The novelty of the paper is not great wrt the previous work, it consists mainly of the use of adversarial learning.
>
> We respectfully disagree with this statement. The novelty of our work is in identifying a new, practically relevant problem (contextual anomaly detection) and adopting existing approaches in a novel way. Indeed this new modelling approach is a fusion of existing techniques, but they haven’t been applied to the time series domain and required non-trivial extensions. We will clarify our contributions better in the paper revision.

---

> > ### Author Response · Authors · 2021-11-23
> > **Follow-up on baseline results on residuals**
> >
> > As promised, we are following up with more baseline results tested on residuals signals:
> >
> > | Dataset \ Methods | Local Outlier Factor |  Isolation Forest |   One-class SVM   | Context invariant embeddings |
> > |:-----------------:|:--------------------:|:-----------------:|:-----------------:|:----------------------------:|
> > |     Synthetic     |   0.537 (+/- 0.046)  | 0.500 (+/- 0.000) | 0.500 (+/- 0.000) |       0.999 (+/- 0.002)      |
> > |      Pendulum     |   0.467 (+/- 0.030)  | 0.500 (+/- 0.000) | 0.500 (+/- 0.000) |       0.980 (+/- 0.002)      |
> > |       DevOps      |   0.573 (+/- 0.070)  | 0.500 (+/- 0.000) | 0.500 (+/- 0.000) |       0.587 (+/- 0.007)      |
> > |      Turbine      |   0.679 (+/- 0.000)  | 0.500 (+/- 0.000) | 0.503 (+/- 0.000) |       0.736 (+/- 0.022)      |
> >
> > It is evident that baseline methods do not profit from residual representations on the synthetic datasets (synthetic, pendulum). However, on both the Turbine and the DevOps data set, LOF achieves solid gains by switching to residual signals. Despite these improvements, context invariant embeddings still outperform these baseline approaches consistently.

---

### Decision · Program_Chairs · 2022-01-20

**Decision:**

Reject

**Comment:**

This is a representation learning time series paper.

The reviewers appreciated aspects of the paper, but all agreed that primarily the experiments are lacking and to a lesser degree the presentation is unclear and needs further proofreading.

So definitely this work has merits. It is also much appreciated that the authors throughout the discussion have been engaged in adding results and further clarification. This can be used for an updated version for the next conference.